# Systemic Barriers and Equitable Interventions to Improve Vegetable and Fruit Intake in Children: Interviews with National Food System Actors

**DOI:** 10.3390/ijerph16081387

**Published:** 2019-04-17

**Authors:** Sarah Gerritsen, Sophia Harré, Boyd Swinburn, David Rees, Ana Renker-Darby, Ann E. Bartos, Wilma E. Waterlander

**Affiliations:** 1School of Population Health, University of Auckland, Auckland 1142, New Zealand; sophia.harre@auckland.ac.nz (S.H.); boyd.swinburn@auckland.ac.nz (B.S.); aren830@aucklanduni.ac.nz (A.R.-D.); 2Synergia Consulting Ltd, Auckland 1011, New Zealand; david.rees@synergia.co.nz; 3School of Environment, University of Auckland, Auckland 1142, New Zealand; a.bartos@auckland.ac.nz; 4Department of Public Health, Amsterdam UMC, University of Amsterdam, 1105AZ Amsterdam, The Netherlands; w.e.waterlander@amc.uva.nl

**Keywords:** child nutrition, systems dynamics, food system, food environment, fruit intake, vegetable intake

## Abstract

Fruit and vegetable (FV) intake is declining in New Zealand, and over half of New Zealand’s children do not meet the recommendation of two serves of fruit and three serves of vegetables daily (with even lower adherence among children in high-deprivation neighbourhoods). The aim of this study was to map the potential causal pathways explaining this decline and possible actions to reverse it. Semi-structured interviews were held in April–May 2018 with 22 national actors from the produce industry, food distribution and retail sector, government, and NGO health organisations. The qualitative systems dynamics method of cognitive mapping was used to explore causal relationships within the food system that result in low FV intake among children. Barriers and solutions identified by participants were analysed using thematic analysis and according to a public health intervention framework. Participants were in agreement with the goal of improving FV intake for health and economic outcomes, and that health promotion strategies had been ineffectual to date due to multiple systemic barriers. Common barriers discussed were poverty, high food prices, low skills/knowledge, unhealthy food environments, climate change, and urbanization. Solutions with the strongest evidence of efficacy identified by the participants were subsidizing FVs and early childhood interventions to improve FV exposure.

## 1. Introduction

Eating fruits and vegetables (FVs) in childhood provides valuable nutrients for growth and development, strengthens immunity, aids digestion, decreases the risk of obesity and obesity-related illnesses, and establishes a healthy dietary pattern for life [1]. Despite this, many children worldwide do not eat a minimum of 400 grams of FVs daily as recommended by the World Health Organization. The annual New Zealand (NZ) Health Survey collects parent/caregiver-reported usual number of serves per day of FVs for children aged 2–14 years. The 2016/17 NZ Health Survey reported that 51% of children had the age-appropriate recommended serves of vegetables a day (two serves for children aged under five years, and three serves for children 5–14 years old), and 72% had the recommended two or more serves of fruit a day [2]. In the last five years, fruit intake has remained unchanged and vegetable intake has declined among children (Figure 1). Vegetable intake is decreasing among children of all ethnicities but is lowest among non-European/Pākeha and in neighbourhoods of high deprivation; 43% of children living in the most deprived neighbourhoods (NZDep deciles 9–10) meet the vegetable intake recommendation [2].

A systematic review of studies on children’s FV intake found age (lower among older children), gender (lower in boys), and socioeconomic position (lower in areas of high deprivation) were strong determinants of FV intake [3]. Personal preference for vegetables and fruit, parental intake, and role-modelling and home availability/accessibility were also found to be related to children’s FV intake [3]. Qualitative studies have suggested that additional determinants of FV intake are the preparation time needed; price and inconsistency of taste; satiety value; appropriate time/occasions/settings for eating FVs; sensory and physical aspects; variety, visibility, methods of preparation; access to unhealthy food; the symbolic value of food for image, gender identity, and social interaction with peers; and short-term outcome expectancies (i.e., parents are not focused on long-term health but immediate concerns) [4]. The majority of research on children’s FV intake focuses on the challenges of increasing consumption at an individual level. There is currently less research that focuses on the impact of structural factors on children’s consumption of FVs, such as price levels, policy, guidelines, supply, and exposure to marketing [4]. Research that takes a wider systems perspective may help to fill this knowledge gap, particularly because food has many different dimensions including health, equity, social and cultural dimensions, and economic and commercial interests. 

The United Nations Children’s Fund (UNICEF) has recently stated that “actors across local, national and global food systems need to be held accountable for providing healthy, affordable and sustainable diets to children and adolescents today and in the future”, particularly because food environments currently do not encourage nutritious diets for children and adolescents, nor are they incentivized to do so [5]. The systems approach in this research makes connections between children’s FV intake, the national food system (e.g., urban growth pressures on fertile-rich land [6], an increasingly globalised food system [7]), and wider social and economic issues (e.g., in New Zealand there is concern about family food poverty [8], which hinders access to and affordability of FVs). This paper takes a systems approach to better understand the factors that may influence children’s FV intake in New Zealand.

The aim of the present study was to identify national food system actors’ views of:
current systemic barriers to meeting the fruit and vegetable (FV) intake guidelines among New Zealand children aged 2–14 years; andacceptable and equitable options for systemic interventions which would improve nutrition by increasing children’s consumption of FVs.

## 2. Materials and Methods

### 2.1. Cognitive Mapping Interviews

Cognitive mapping interviewing (CMI), sometimes referred to as causal mapping, was used to collect qualitative data during the interviews and assist with data analysis on barriers and solutions. CMI is a visual technique developed by Eden (1988) [9] that captures a stakeholder’s perspective of a problem or issue and how they or their organisation relate to the issue. CMI is one tool used in systems research, which views issues, such as low FV consumption in children, as emerging from complex and adaptive systems [10]. The CMI method is underpinned by the psychological Personal Construct Theory [11], which asserts that individuals interpret events in unique ways but can develop a shared understanding or interpretation, especially when based on common goals/outcomes. A set process is followed during a semi-structured interview to order and reflect the interviewee’s view of the social system. The process of creating the CMI map aids participants’ reflection and sense-making [12]. The map produced during each interview consists of a network of nodes and arrows as links that are usually linear in structure [13]. The individual maps from each interview can then be combined to form a composite map, presenting the subjective data in a diagrammatic form that gives a holistic and systemic view of an issue. CMI has most often been applied to complex management problems [12] and in operational or action research [14]. This project is one of the first to use CMI in public health research. CMI were used in a health services project to develop a theoretical framework for implementing innovative practices in primary healthcare management in New Zealand [15], and other studies internationally have used similar system dynamics methods [10,16,17,18,19]. 

### 2.2. Procedures

Participants for CMI were identified through the researchers’ existing networks or from publicly available information on websites. Criteria for participation was based on employment in a prominent role in one of four sectors of the New Zealand food system: government/regulatory, produce industry, retail industry, or public health/nutrition promotion. Attempts were made to include at least two participants from each of the four sectors, including someone from both of the two main supermarket chains in New Zealand. Seventeen individuals from different organisations were invited by email in March 2018 to take part in a one-hour interview about declining FV intake in children. Twelve agreed to be interviewed, and the other five sent the invitation on to colleagues within their organization whom they believed would better represent the organisation on this issue. Four of these people agreed to take part. If there was no response to the email, a researcher phoned to follow up. Participants were given the choice of time, day, and location for the interview. Some organisations requested that multiple people within their organisation attend the interview to cover different areas of expertise.

All participants provided written informed consent for inclusion before they participated in the study. The study was conducted in accordance with the Declaration of Helsinki and approved by the University of Auckland Human Participants Ethics Committee on 1 March 2018 (Reference: 020684). 

The interviews began with participants being shown a graph of New Zealand Health Survey data showing the proportion of children aged 2–14 years who meet the FV intake guidelines from 2012/2013 to 2016/2017 (Figure 1). The interview schedule then contained six questions:
We can see (in the graph) that in the past 5 years, fruit intake hasn’t changed and vegetable intake has declined overall, with a pronounced drop in 2015. What do you consider to be the primary causes of these patterns? What do you think are the consequences if these patterns continue? What would be the consequence if these patterns changed and increasing numbers of children consumed the recommended amount of fruit and vegetables?What would need to happen to bring these lines up to 100%?Fruit and vegetable intake is lowest among children who live in low-income communities. Why do you think this is, and would they require different interventions to see these patterns change?Is there anything else you would like to add on this issue before we finish?

Interviews were audiotaped and transcribed verbatim. During the interview, the researcher constructed a hand-drawn cognitive map, which was referred to during the interview to check the interviewer’s interpretation of comments. The interview transcript, map, and consent form were linked by participant number, rather than name, to maintain confidentiality. Participants were given a small gift (a bottle of local olive oil) at the conclusion of the interview to thank them for their involvement. Following the interview, participants were emailed the transcript and a refined version of the cognitive map constructed during their interview and were given two weeks to modify, delete, or correct any of the information. Quotes from the interviews have been used in this manuscript to provide examples and elaborate on the main findings in the results section. 

### 2.3. Analysis

Variables listed on each cognitive map were entered into a spreadsheet for thematic analysis. Variables were grouped under three headings: “barriers”, “solutions”, and “outcomes”, with similarly worded variables combined. Quotes in relation to the key variables were extracted from the transcripts. Variables were printed off onto single strips of paper and three researchers worked together to place each variable into subthemes before grouping subthemes into broader main themes. These subthemes were recorded on the spreadsheet and colour-coded by sector to analyse differences between and within sectors. A composite summary map, combining the maps from different interviews, was created with Vensim software using the most commonly mentioned variables on the spreadsheet (mentioned in four or more interviews). In accordance with the CMI method, the structure of a composite map is a triangular pattern with “system goals” at the top of the triangle, “strategic directions” in the middle, and the “potential options” at the bottom of the triangle [13]. Variables are worded to maintain a neutral position on the map, and arrows illustrate if one variable increases or decreases the variables it is linked to. Each vertical linkage of variables tells a “story” about the relationship between goals, strategic directions, and potential options/solutions.

In order to interpret whether the solutions proposed by participants would have an equitable result, the Public Health Interventions Framework developed by Adams et al. (2016) [20], which consists of two scales on tangential axes, was used. The vertical *y*-axis represents where the intervention is targeted, from “population” to “high risk” individuals, families, or communities. The horizontal *x*-axis indicates the level of individual agency required to benefit from an intervention, from low to high agency. According to Adams et al. (2016), interventions that are placed as “high risk” and “low agency” are likely to be the most equitable and most effective. The solutions proposed by participants during the interviews were extracted from the spreadsheet and placed in the appropriate position on a diagram of the framework if they contained enough detail to be considered as a potential intervention. 

## 3. Results

### 3.1. Description of Interview Participants

Table 1 contains details of each interview and the participants’ roles and organisations. Only one of the 17 organisations approached (a large supermarket retailer) did not participate. Most interviews were conducted in-person at the interviewee’s workplace, but two were held at the University of Auckland and two were by videoconference. Three interviews contained more than one participant from the same organisation (112, 113, 115). At these interviews, each participant was given the opportunity to answer every question and one cognitive map was produced for the interview.

### 3.2. Themes from the Interview Maps and the Composite Summary Map

Sixteen individual CMI maps were produced during the interviews (attached as Appendix A). Saturation in the barriers and interventions proposed was reached by the 14th interview.

Originally, 362 variables were extracted from the 16 individual CMI maps (including barriers, solutions, and outcomes). Thematic analysis of the variables began with the consolidation of similarly worded variables, resulting in a final total of 240 variables. Barriers and solutions were grouped into 26 subthemes and eight main themes (Table 2). 

Figure 2 presents the most commonly mentioned variables during the interviews, ordered using the CMI method, which shows horizontally from the top: the system goals (outcomes), strategic directions (which includes barriers), and then drivers of the system, and presents the stories told by participants vertically. 

### 3.3. Consequences of Low Fruit and Vegetable Intake in Children

All participants identified multiple economic and health-related impacts of low FV intake in children. Poor health outcomes were identified in all but one interview, with participants referring to health as physical and/or mental. Physical health outcomes included chronic and non-communicable diseases such as diabetes and obesity. Participants linked mental health to moods, motivation, and productivity. Three interviewees suggested that low FV intake is linked to a widespread culture of poor nutrition, which, in turn, promotes poor health outcomes. Two interviewees suggested that poor health outcomes would be differentially spread, contributing to health inequalities. The consequences of improving FV intake in children were viewed positively. In two interviews, participants suggested that an increase in FV intake would lead to children and adults working and studying at a higher capacity, thereby reducing social inequalities. 


*“Lack of vegetable intake, in particular, is probably like a proxy for diet poverty as a whole. It’s obvious the nutrients that vegetables provide, but looking wider than that, lots of processed foods (create) micronutrient deficiencies in children, and their ability to grow. There’s a knock-on effect too, oral health… chronic disease is that end-point, but that middle point in between is a less healthy group of children and adolescents that have poorer–not just health outcomes, but educational outcomes too.”*
*—Interview 112*


*“We already have really high rates of non-communicable diseases like obesity, and diabetes, and malnutrition. And I think that those are just going to escalate. And those children are going to grow up to have children who will follow the same habits and are also just as likely to be in maybe low-income areas if they already were in low-income areas. I think the problem will escalate. Absolutely.”*
*—Interview 109*

The economic impact on the health system due to increased poor health was mentioned in four interviews. Two interviewees stated that low FV intake will have economic impacts in rural areas, on commercial businesses, and on growers. Several participants worried that low FV intake would lead to grower consolidation and one grower said low FV intake could eventually lead to fewer sales. However, other participants representing growers and supermarkets noted that they are seeing increased sales of FVs despite a trend of decreasing intake in children. While individual growers raised concerns around their economic livelihood, large-scale growers and supermarkets did not feel their businesses were threatened by low FV intake and felt insulated from the economic impacts.


*“In some ways, it’s a fairly fragmented industry. What we hear from our growers, which are generally large farming operations, may not be the overall sentiment of the grower community. We’re selling plenty. They’re selling plenty. We don’t see what’s going on in the whole market... I’d imagine the smaller, less efficient growers would be most impacted by a reduction in volume…”*
*—Interview 104*


*“If you have a look, there’s massive consolidation in the grower base in New Zealand. When I started 30 years ago in the industry, there were 60, 70 cauliflower growers in Pukekohe. Today there’s six. And there has had to be consolidation so that they can survive and become efficient. It’s all about volume. But you get to the point where you can’t be any more efficient.”*
*—Interview 105*

### 3.4. Barriers to Fruit and Vegetable Intake in Children

The most commonly mentioned barriers to children eating FVs were: negative perceptions of the price of FVs; actual price of FVs; urbanization; climate change; living in low-income households and poverty; having a low food budget; lack of time for preparation; eating unhealthy replacement foods; lack of knowledge of eating seasonally; natural preferences for sweet tastes over bitter; and knowledge and skills lost through generations. 

#### 3.4.1. Fruit and Vegetable Prices

High FV prices were mentioned in 10 of the 16 interviews, and by participants in all sectors. However, the reasons given for the high FV prices varied, and there was disagreement about whether FV prices were increasing or whether this was driven by perceptions rather than actual high FV prices. Some participants stated that fewer imports of vegetables (compared to fruit) kept fruit prices more stable than vegetables. While most participants attributed the high prices to extreme and unpredictable weather events, one participant suggested that supermarket margins were to blame. 


*“We’ve had lots of strong rains and storms coming through, and I’ve read about rotting kumara in the ground, rather than it getting to be dug up and sold, therefore, driving up the cost. Things like cauliflower; that’s been crazy.”*
*—Interview 107*


*“One of the things that really, really worries me that I think could help, is that retailers don’t make any money in grocery. It’s all about shelf space, volume, and small margins. Fresh food is where retailers make their money… in (grocery) it’s all about a price point. They pretty much do it at either cost or a small margin. Coca-Cola or Arnott’s or Greg’s or whoever will buy the shelf. The middle shelves at eye level or at the end of the aisle, the supermarkets will get thousands and thousands of dollars for it. They make their money out of selling shelf space, not the product themselves. In fresh foods that’s where they make their margins. Produce makes up about 10% of the store’s turnover but probably close to 25% to 30% of their margin. I have no issues with the margin that retailers make. I mean, they’re running big businesses… it just seems a little bit back to front that all the stuff that’ll kill you is cheap and the supermarkets don’t make their margin on it. Versus all the stuff that’s healthy and fresh is where they make their margin.”*
*—Interview 105*

Four participants said that FV prices were only perceived to be high, and the actual prices were not a barrier. Two participants attributed this perception to increased media attention and publicity surrounding Consumer Price Index (Statistics New Zealand) data on rising FV prices and less attention on the simultaneously rising costs of unhealthy food products. 


*“There’s been a lot of press about fruit and fresh veg prices, and there hasn’t been anybody jumping up and down about increasing prices of fast food, have they? I tend to go 2.5 kilos of potatoes at 8 bucks versus other vegetables or other things for a kilo, there might’ve been a 5% increase and it’s way out in front, but it’s still bloody good value compared to other things.”*
*—Interview 115*


*“Everybody’s been saying price... And I think at that time, yeah, sure, cauliflower was $10 or whatever it was, but you could still buy lettuce for 99 cents… When people are talking about the price, they’re not talking about if it’s old-season, new-season, the quality or anything like that.”*
*—Interview 116*

#### 3.4.2. Limited Household Food Budgets

Four interviews linked less access to FVs with poverty and low-income families’ inability to stretch the dollar to include fresh food in their food budgets. Low income and poverty was linked to less access to FVs and low family food budgets, which reduced the ability to purchase FVs.


*“If you look back at the socioeconomic deprivation over the recent years and housing costs, that although our employment has been fairly good, the difficulty of bringing up a family on current incomes where there hasn’t been much of a move while costs have gone up and up, I would think that financial stress plays a huge part”*
*—Interview 103*


*“The reason why [we have low FV intake in children] is because of families and children living in low-income situations and living in poverty. Knowing that food is generally the last item that people buy, so that’s when they’ve got the pay packet each week, whether it’s from benefits, or paid work, or tax credits, it pays the rent, pays the bills, pays the outstanding debt. Food is the last items that you choose. When I go to the supermarket and look at fruit, how much you pay and what you get, I can see why families would choose to get high-energy food, and easier foods to prepare.”*
*—Interview 101*

#### 3.4.3. Advertising and Promotion of Unhealthy Food

Most participants mentioned barriers to children eating FVs within the food environment, including home, school, and local retail environments, as well as marketing and advertising. Many participants referred to the advertisement and promotion of unhealthy food, including event sponsorship, and some participants noted that there was little marketing of FVs to children in New Zealand. 


*“Marketing and advertising contributes to children asking for packaged food. The food environment is cluttered with mixed messages, and where nutrition literacy is lower, parents and caregivers are unsure of what to give their children. The perceived quick and easy option is processed food.”*
*—Interview 110*


*“The other thing I was thinking of was looking at policies around advertising and sponsorship. You see those big sponsors still sponsoring huge events, like Coca-Cola Christmas in the Park. I don’t know if we can even match that, but that’s what we’re up against.”*
*—Interview 111*

#### 3.4.4. Children’s Food Preferences

Some participants also mentioned food preferences and taste, noting that children have a natural preference for sweet tastes over bitter tastes and a preference for unhealthy food. 


*“It’s been known for a long time that children have a preference for things that are slightly sweeter so apart from the fact that the number of serves required of fruit a day is lower than that of vegetables… most kids prefer the flavour of fruit to vegetables because it’s sweeter.”*
*—Interview 113*

#### 3.4.5. Limited Availability of Fresh Produce

Within the retail environment, the most commonly mentioned barriers were the limited availability of FVs at dairies (corner stores) and the location and density of unhealthy food outlets, particularly in areas of high deprivation.


*“Thinking of places like (suburb), they don’t have a supermarket anymore due to it being earthquake-prone and for whatever reasons, it has financially not been viable. Whereas, there’s a laundromat, and a dairy, and a couple of hot roast shops and things like that. The dairy might have a few manky old bits of fruit, but there’s not anywhere for them to go in walking distance to get fresh fruit and vegetables.”*
*—Interview 107*


*“You go to (suburb) and look in the mall there… every second shop is a fast food takeaway. And very little in the way of fresh food. But you’ve also got the advertising which promotes junk food, sugary drinks, all that sort of thing everywhere.”*
*—Interview 103*

#### 3.4.6. Lack of Time for Food Preparation and Skills/Knowledge

Participants also noted that many parents have a lack of time to purchase and prepare vegetables for eating, which is compounded by work commitments and the need for two-income households. Participants suggested that a lack of time also led to parents purchasing processed and convenient food options, which were often unhealthy and contained limited FVs. One participant stated that time was more of a barrier for low-income families. 


*“The misconception that we hear is that people living in poverty or people on low incomes have lots of time. But that’s not true, especially if you’re in the welfare system, you’ve got all these appointments to go to. If you’ve got children, you need to look after your children. That involves time. And you may have other family members that you’re looking after too.”*
*—Interview 101*


*“More and more mums are working full time now. They’re not in the house to prepare meals. It’s like everybody gets home at 5, 6 o’clock. Everybody’s tired. Everybody wants to eat quickly. You just don’t have time to do it.”*
*—Interview 112*


*“Although fruit and vegetables are quite convenient, there’s an idea that takeaway foods are more convenient, and you wouldn’t be getting lots of vegetables (in takeaways) unless you chose very carefully.”*
*—Interview 106*

Participants from most interviews linked the purchasing of processed and convenient foods to limited cooking skills and/or a lack of nutrition knowledge. This was associated with the idea that knowledge and skills have been lost through generations, and that there has been a removal of “home economics” and cooking from the school curriculum.


*“Who is there to teach our kids to cook? My grandma and my mum taught me how to cook. But my mum has been working; I don’t have time, because I work full time and I’m a single mum, so my kids can’t cook. And my sons at the age where he’s ready to move out, and he’s lucky if he can make himself a toasted sandwich, and that’s on me. That’s on me. With a population that can’t cook for themselves, and can’t fend for themselves, what’s their alternative? Junk food.”*
*—Interview 112*

#### 3.4.7. Loss of Fertile Land to Urban Development

Barriers participants mentioned about the built and natural environment included urbanization and the impact of climate change on growing conditions. Urban sprawl was identified as a barrier to commercial FV production and increased neighbourhood density a barrier for growing home gardens.


*“Unfortunately, a lot of development tends to be on productive soils because it’s flat. I don’t know if you’ve looked at future urban area from Westgate out to Kumeu (North-West Auckland). That’s all designated future urban out there, and that’s where all the strawberry farms are. The strawberries only grow in class 1 soils, and that’s all going to be built over. And 98% of the class 1 soils are in South Auckland, so that means not much of it is in Rodney. The majority of that will be gone once that Westgate to Kumeu goes through.”*
*—Interview 109*


*“There is also the urban sprawl going on, so that’s taking away some of the land as well. The growers might give up (and) sell for urban sprawl. There’s a grower on one corner and he’s getting hemmed in with houses. And (another grower) sold all that land which is now developed with roads, no houses, just roads, for 10 million. And of course, that’s really tempting.”*
*—Interview 102*

#### 3.4.8. Government Regulations

Three interviewees thought increasing government regulation produced barriers for growers and distributors, noting particularly that the Ministry of Primary Industry restricted imports of some FVs (due to biosecurity concerns), which limited supply. Others noted that the Good Agricultural Practice (GAP) certification was much more rigorous than government regulation. During their interviews, government officials stated that although there were some changes to the paperwork required following the new Food Act, there was minimal regulation of fruit and vegetable growers overall. 


*“The businesses that trade food have to comply with the Food Act, 2014. The Food Act came into effect in March 2016 and it’s a three-year rollout. And, in fact, we’re only just getting to the horticulture sector. But we’ve been doing quite a bit of work with the horticulture sector to try and make it as straightforward as possible. Broadly, up until now, from a sort of food safety perspective, growing fruit and veg hasn’t fallen under food and safety legislation, but it’s about to. There’s a registration process for a business to go through. However, it’s pretty minimal. If you’re a packhouse, for example, if you’re growing things or packing them, there’s a registration, there’s a check that you’re meeting the requirements of the Act. And after that, there’s no future programme for checks. So you would only have somebody coming back if something went wrong.”*
*—Interview 113*


*“There’s a lot more paperwork for growers these days and a lot more regulation, a lot more rules, a lot more laws that they have to abide by. What you’re seeing is smaller growers just giving up. And you’ve got much bigger growing operations now. And some of those operations will have multi-sites through the country, which cushions some of the weather, depending on what they’re growing.”*
*—Interview 102*

### 3.5. Solutions to Low Fruit and Vegetable Intake in Children

Participants identified fewer solutions compared to barriers within their interviews. Often, specific interventions were not discussed in detail but most participants expressed a need for multiple actions throughout the food system. The most commonly mentioned solution was increasing children’s exposure to FVs through school-based programmes, followed by increasing knowledge of how to purchase and prepare seasonal FVs. Other commonly mentioned solutions included removing the goods and services tax (GST) (currently, 15% is applied to nearly all goods and services in New Zealand, including imported products) from FVs; instore promotions of FVs; and increasing knowledge of the health benefits of FV. 

#### 3.5.1. Health Promotion in Schools

The majority of interventions proposed by participants focused on changing the school environment. Ten out of the sixteen participants thought that increasing exposure to FVs through school-based programmes would increase children’s FV consumption. In six interviews, it was suggested that nutrition education should be included in the school curriculum. Participants also said that schools should have healthier environments. Most participants spoke positively about Fruit in Schools. Fruit in Schools is a Ministry of Health-funded programme to provide a free piece of fresh produce (fruit or vegetable) to children daily in targeted primary schools (Years 1–6/8). Currently, the programme is in every decile 1 and 2 school (areas of high deprivation). In five interviews, participants thought the Fruit in Schools programme should be extended to include all schools, or a wider range of schools, and/or more vegetables. 

#### 3.5.2. Educational Interventions

Participants identified opportunities to increase parents’ knowledge about how to purchase and eat seasonal FVs within the home and retail environment through health and in-store promotions. Although most participants saw a role for health promotion, they were critical of efforts so far. Several participants mentioned the longevity and high awareness of the 5+ A Day campaign. 5+ A Day is a charitable organization that develops curriculum-linked resources promoting awareness, understanding, and knowledge of the importance of eating 5+ A Day, which are produced and distributed free to educators and health professionals. However, most participants felt this health promotion programme was not effective due to environmental barriers. 


*“The problem is the kids are not able to action those [healthy eating] messages that they are getting because of their environment, whether it be home, or the supermarket, or the dairy, or whatever. So they know that they should be eating their fruit and veg but actually being able to do it is totally different.”*
*—Interview 107*


*“One of my first thoughts was the 5+ A Day campaign is a failure, because of the data you’ve just presented. If the 5+ A Day campaign was working, you’d expect to see an improvement rather than a decline… One thing that I used to challenge 5+ A Day on was you don’t go right across the store. The Health Star Rating will go right across the store. If you think about a retailer’s desire to get behind something, I’m pretty sure a marketing team would rather market something that goes right across the store.”*
*—Interview 104*

#### 3.5.3. Price Promotions

In-store promotion of FVs was mentioned in five interviews, and the participant representing the retail sector stated that improving in-store promotion would lead to increased sales. This included presenting growers’ stories, improving lighting, seasonal specials, and drawing attention to “good buys”. Several participants felt the government could provide funding to support initiatives aimed at increasing FV consumption or sales, with one participant mentioning that a government-funded campaign “celebrating seasonality” would be helpful. Some participants stated that if parents understood seasonality then they could better cope with price fluctuations of FVs.

#### 3.5.4. Government-Led Interventions

Potential interventions for government to lead were mentioned in most interviews. This applied to the above solutions relating to the school, home, and retail environments and extended to new legislation for council zoning of food outlets and removing GST from FVs.


*“Planning and consenting around land use is quite important. Consenting would be around the types of food businesses that can locate in certain areas, or the numbers of them, concentration of fast-food stores, for instance, in a certain area, or proximity to schools, for example.”*
*—Interview 109*


*“Well, if removing the GST makes enough of a price difference (I’d support that)… I know that’s going to be considered. And we’re obviously supporting anything that makes fruit and vegetables more accessible.”*
*—Interview 108*

Although several participants mentioned GST removal as an intervention, they were unsure about the implementation of this policy and if it would be effective. 


*“I don’t know where I personally stand on subsidising fruit and vegetable because I have read both arguments of people not wanting to change the GST from the 15% on everything. But it has been done in some places internationally. And it has had mixed results. But I can see a case for it. How it would pan out in a New Zealand environment, I don’t know. I haven’t looked into it enough.”*
*—Interview 107*


*“I don’t really know about the evidence around reducing the costs of GST on fruits and vegetables and things like that. Will that actually translate to families buying them more? Because I think it still comes down to food skills, and preferences, and time, and all those other barriers.”*
*—Interview 112*

#### 3.5.5. Early Interventions

Several participants mentioned the importance of early intervention. This included repeated exposure to unaccustomed tastes from an early age and creating vegetable-focused guidelines for early childhood centres (ECEs) and schools.


*“But I think if you’re starting right from that early age, and embedding it within children as a priority, and that is what is normal, I think that’s probably—for me, personally, I think it’s a good investment.”*
*—Interview 112*


*“You start Year 7 with cooking at school… The horse has bolted by the time they’re Year 7 and 8”*
*—Interview 118*

### 3.6. Differences by Sector

Some small differences by sector were prevalent between the interviews. However, across the barriers and solutions there was a lot of agreement, and there was significant convergence on the “system goals” around the importance of improving children’s FV intake. The participant representing the retail sector identified solutions derived by corporate social responsibility (CSR) as a way forward. This included in-store promotion and investment in the produce department. Although most participants saw a role for government and supermarkets in promoting FV intake to children and their parents, growers saw their role in supporting FVs through industry-funded programmes (e.g., vegetables.co.nz) as important too.

### 3.7. Equity and Effectiveness of the Interventions Proposed

The solutions mentioned by participants were predominantly targeted at the general population rather than high-risk individuals, and many required a high level of individual agency (refer to Figure 3). Extending the *Fruit in Schools* programme was the most commonly mentioned solution targeting those at high risk of developing diet-related diseases and requiring a low level of individual agency to benefit. Increasing welfare benefits to increase food budgets and lunch programmes in schools are also in the same quadrant of the public health interventions framework (i.e., top left of Figure 3). Council zoning of fast-food stores requires low individual agency and sits in the middle of the target area axis as it could be applied to either the general population or implemented only in areas of high deprivation. 

## 4. Discussion

This research explored the systemic barriers to children eating FVs and gathered ideas for potential interventions to reverse the decline in FV intake in NZ. The summary cognitive map (Figure 2) combines the individual cognitive maps created during the interviews and shows the common themes mentioned by the research participants. The actors from across all parts of the food system held similar views about the importance of children eating FVs, believing that declining FV intake would result in poor health outcomes, increased costs to the health system, and ultimately have a negative effect on the NZ economy.

While the most commonly discussed barriers were structural issues such as poverty, high FV prices, and the changing natural environment, some participants focused on individual behaviour change interventions, such as increasing knowledge about eating seasonal FVs. This lack of alignment showed that even though most participants could identify structural barriers to children eating FVs, some were unable or unwilling to suggest structural interventions to target those barriers. Adams et al. (2016) and other public health researchers [21,22] have suggested that interventions which require high individual agency are often more palatable for the public, politicians, and the private sector than low agency interventions that target structural barriers [20]. Personal solutions to structural problems reinforce the predominant neoliberal political economy of the food system and may be more likely to reinforce socioeconomic inequalities in health [23]. Furthermore, exerting personal agency requires cognitive, psychological, temporal, and material resources, all of which are largely patterned by socioeconomic status and privilege [20].

Nevertheless, interventions which fall across the public health interventions framework were suggested by the research participants (Figure 3), and a combination of these could be explored and tested further. Before implementation, each proposed intervention would need to weigh the health benefits to individuals and society against the potential erosion of individual freedom (if choice is reduced or other restrictions occur) alongside efficacy, economic cost, and benefit to equitable population health outcomes [24]. Some of the most commonly suggested interventions proposed by participants are now considered in detail, along with evidence from the literature about their efficacy and effect on equity.

### 4.1. School Environment

The most commonly mentioned solutions to low FV intake in children proposed by the national actors in this research centred on the school environment. Most participants suggested that nutrition education should play a larger part in the school curriculum and that children should have increased exposure to FVs through school programmes. In NZ, there is varying nutrition education between schools [7]. Several health promotion programmes operate in NZ schools, more commonly found in areas of high neighbourhood deprivation (Heart Foundation, personal communication). As mentioned earlier, the *Fruit in Schools* programme is run by the Ministry of Health and offers FVs to all children in the most economically deprived neighbourhoods (deciles 1 and 2) [8]. Other programmes are administered by charities or businesses, such as gardening and cooking programmes (e.g., *Garden to Table* and *Kids Can Cook*), lunch provision such as *Eat My Lunch* and *KidsCan*, and promotion of FVs such as 5+ A Day. One participant observed that schools in areas of high deprivation were saturated with nutrition programmes and that adding more would not improve children’s FV consumption. 

An international review and meta-analysis of 27 school-based FV programmes for 5–12 year olds estimated that they improved daily FV consumption by an average of one-quarter to one-third of a portion (the equivalent to a 20–30 g daily increase) [25]. Even though nearly all of the programmes aimed to increase FV intake, most only achieved an increase in fruit consumption and not vegetables. Multicomponent programmes that motivated and engaged children and families to change their eating behaviours at home tended to result in larger improvements in FV intake, but these programmes were difficult to replicate without considerable time, manpower, and funds. Single-component programmes that provided and distributed free or subsidized FVs were less effective but more highly rated by school staff and teachers as they were easier to implement than multicomponent interventions. The systematic review concluded that school-based programmes, including distribution schemes, have the potential to moderately improve the daily consumption of fruit, but more work is needed to design programs to improve vegetable intake in children and to reduce barriers to positive behaviour change [25].

There is also some limited quantitative evidence for the positive effect of school gardens on children’s nutrition, particularly regarding increased preference for and consumption of FVs [26]. In comparison to standard curriculum-based nutrition education (for example, a lesson in the classroom on the nutrients in fruits) experiential learning in a school garden has been found to have a greater influence on children’s consumption/energy intake and on increasing nutritional knowledge [27]. Langellotto and Gupta’s (2012) meta-analysis found that participation in nutrition education led to an increase in nutrition knowledge. Positive attitudinal (e.g., increased preference for fruit and vegetables) and behavioural changes (e.g., increased consumption) were primarily documented in gardening programmes. They propose that gardening increases access to healthy foods, particularly vegetables, while decreasing the reluctance children may have to try novel foods [28].

Improving the vegetable component (by perhaps supplying both vegetables and fruit daily) in the *Fruit in Schools* programme, as suggested by many of the national actors interviewed, may be one way to utilize the school environment while not oversaturating schools with new health promotion programmes. While the literature shows that school vegetable provision is unlikely to increase consumption overall [25], it may encourage preference for the taste of vegetables and the normalisation of vegetables as a snack, if this was not established in early childhood. 

### 4.2. Repeated Exposure to Fruit and Vegetables in Early Childhood

There is a growing body of literature confirming the importance of repeated exposure to novel and bitter foods in infancy for establishing healthy eating behaviours [26]. Two systematic reviews have concluded that multi-component interventions in preschool settings that incorporate a parent or family as part of the programme provide the strongest evidence for improving fruit and vegetable intakes [29,30]. When considering the intervention setting, longer-term interventions within a preschool setting, as opposed to schools, were most likely to be effective at increasing children’s vegetable intake six months after the intervention. Intervention effectiveness was associated with the number of settings targeted and the frequency of contact, but not the length of the intervention [29]. 

A systematic review of interventions associated with increasing vegetable consumption in 2–12 year old children found that overcoming an aversion to the bitter and unfamiliar textures of vegetables requires repeated exposure and long-term positive reinforcement in order for children to begin to like such foods [29]. Studies of early exposure to specific tastes (sweet, salty, bitter, umami, and sour) and acceptance of those same tastes later in life have concluded that sweet tastes are most probably innately preferred by children [31]. However, exposure to bitter tastes through the use of a formula increased the intake of bitter flavours, supporting the theory that the liking of bitter tastes (such as broccoli and cauliflower) is learnable if introduced early and if mothers consume these foods in pregnancy and while breastfeeding [31]. A randomised controlled trial in the Netherlands suggested that early weaning exclusively with vegetables resulted in a higher daily vegetable consumption until at least 12 months of age, with further research required into the long-term maintenance of this effect [32]. 

### 4.3. The Price of Fruit and Vegetables and Household Poverty

High FV prices was mentioned by nearly every participant. However, many participants suggested that FVs were only perceived to be expensive. Vegetable prices, particularly, do fluctuate according to seasonality, weather, and the variety of crops produced, all of which can affect the supply and availability of fresh produce. Statistics New Zealand’s Food Price Index reported the largest annual increase in vegetable prices during 2017 (e.g., a 500 g head of lettuce was $5.28 in May 2017, compared with $2.12 in May 2016 [33]), but fruit prices have been relatively stable since 2015, and vegetable prices have now returned to their usual seasonal fluctuations [34]. A comparison of fruit and vegetable prices at different outlets throughout NZ in 2013 found “other FV markets” (excluding growers markets) were substantially cheaper than supermarkets [35]. 

Some adults in New Zealand struggle to afford sufficient food for their households at any price point [36]. Lack of food continues to be the main reason for government hardship assistance grants, of which 320,000 were provided last year, up from 190,000 five years ago [37]. The Salvation Army [38] and the Auckland City Mission [39] report increased numbers of families obtaining charitable food parcels, and low-decile primary schools are providing breakfast and lunch to a record number of children [40]. The number of children hospitalised for malnutrition in NZ is low (around 100/year in 2014–2016), however many children admitted to hospital have a micronutrient deficiency and/or are overweight, suggesting that their diet, and particularly FV intake, is inadequate [41]. 

Some participants suggested that removing GST (of 15%) on FVs would enable low-income families to purchase more FVs, thereby increasing children’s exposure to FVs. Several systematic reviews have found strong evidence that subsidizing FVs significantly increases consumption [42,43,44,45,46]. The most recent pooled analyses of 23 interventional and seven prospective cohort studies with 37 intervention arms found that a 10% reduction in the cost of fruit and vegetables would result in a 14% increase in consumption [42]. Reductions in the price of fruits and vegetables reduced body mass index by 0.04 kg/m^2^ per 10% price decrease [42]. Although raising welfare benefits or child tax credits would also assist low-income families to afford more FVs, this policy intervention was only mentioned by one participant.

### 4.4. The Food Environment

Participants mentioned that the high density of unhealthy food outlets, particularly in low socioeconomic areas, was contributing to declining FV consumption in children. This observation is supported by national studies showing that the relative density of unhealthy food outlets was significantly higher in the most deprived areas of New Zealand in comparison to the least deprived areas [47,48]. Furthermore, data also show that there are significantly more convenience stores and fast food and takeaway outlets around the most deprived urban schools in comparison to the least deprived schools [48]. There is currently no central or local government regulation of fast food outlets in school zones or in high-deprivation communities in New Zealand. 

Participants mentioned that population growth and urban sprawl has led to less land for commercial growing. Urban expansion into highly fertile areas has accelerated in the Auckland region since 1996 [6]. Similar to our findings, earlier interviews with vegetable growers in Pukekohe, South Auckland, discussed the temptation of potential financial gains that could be made by selling their farmland for residential development [49]. The new Auckland Plan 2050 attempts to resolve this problem by encouraging growth within rural towns rather than the countryside and also provides some innovative ideas for increasing opportunities for growing food within the city itself [50]. 

### 4.5. Strengths and Limitations of the Study

As far as we are aware, this research is the first public health project to use the system dynamics method of cognitive mapping interviews. A broad range of participants for this research was achieved, however, it would have been preferable to include both major NZ supermarket chains and an independent FV retailer, as well as processed food manufacturers. Drawing the cognitive maps while interviewing participants proved an effective way of engaging the interviewees and assisted them in visualizing barriers and suggesting where connections between variables could be made. It also helped the interviewer to return to topics/variables that were mentioned but not described in sufficient detail during the interview. The usual system dynamics method for qualitatively exploring a complex issue is Group Model Building (GMB) [51]. Cognitive mapping interviews were chosen instead of GMB because the national actors were particularly difficult to schedule a meeting with, so flexibility in the location, time, and date of the interview was required. Additionally, it is unlikely that the “elite” participants would have shared openly if working in a group setting, with rival organisations and businesses present. GMB is a more collaborative method for exploring the barriers and potential solutions to FV intake (as all participants are in the room together) and, arguably, would have made it easier for participants to explore points for interventions that would “disrupt the system” since the map produced is circular, rather than linear as in this study. Cognitive mapping would be an appropriate method for future system dynamics research needing to recruit diverse participants who have time constraints. 

## 5. Conclusions

This project synthesized knowledge from actors across the national food system in New Zealand, namely growers, the produce industry, distributors, supermarket retailers, government policy makers and regulators, and health promoters, to identify systemic barriers and potential solutions to low fruit and vegetable consumption in children. The solutions to the issue of low FV intake in children suggested by multiple research participants and which hold the most promise to address health inequities (as they require low individual agency) were extending the *Fruit in Schools* programme to more schools and/or by providing vegetables to children daily; removing GST (or introducing other subsidies) to reduce the cost of FVs; placing more of an emphasis on FVs and nutrition in the early childhood and school curriculum; increasing children’s cooking skills and exposure to FVs through early childhood education and school-based programmes; introducing regulations to ensure healthy food environments in schools. Subsidizing FVs and early childhood FV interventions that improve exposure to influence taste preferences have the strongest evidence of efficacy in the international literature and should be considered by the NZ government. Using a systems approach to research the declining FV intake trend in children has provided new insights into this complex public health issue.

## Figures and Tables

**Figure 1 ijerph-16-01387-f001:**
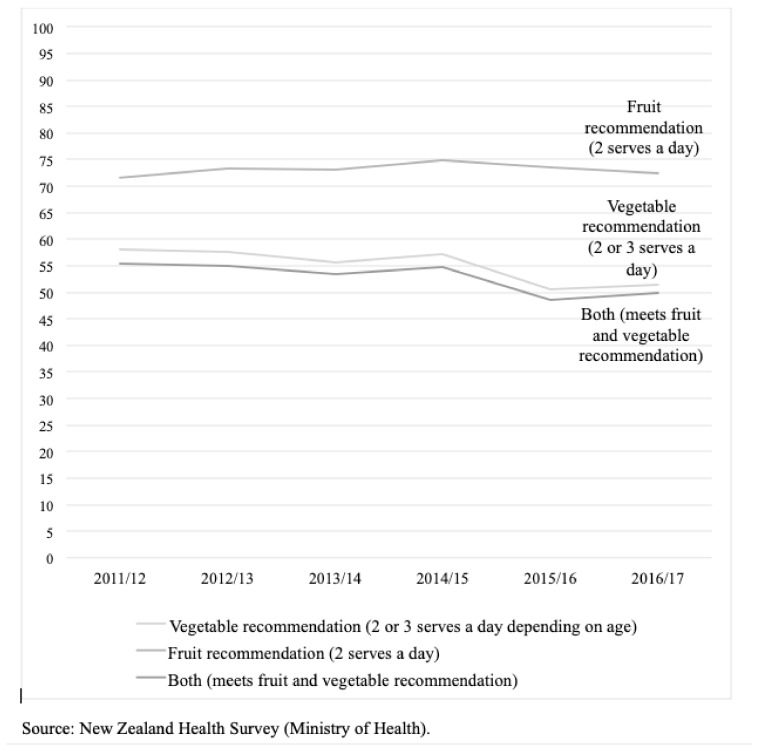
Proportion of children aged 2–14 years eating two serves of fruit and three serves of vegetables daily, 2012/2013–2016/2017 [2].

**Figure 2 ijerph-16-01387-f002:**
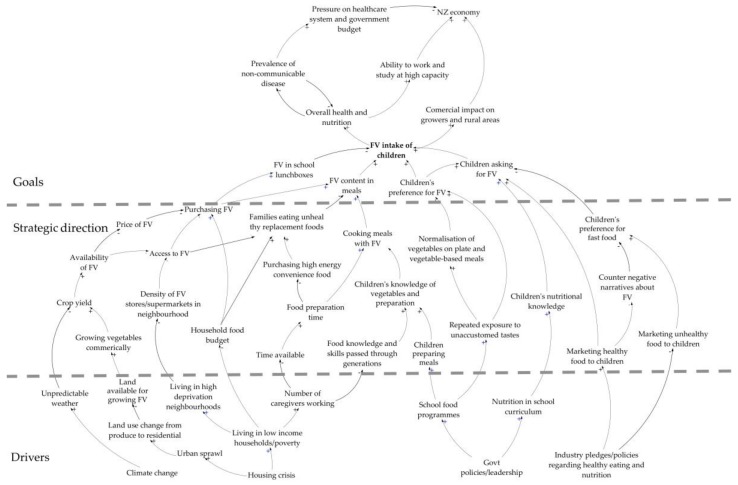
Composite summary map from interviews with national food system actors.

**Figure 3 ijerph-16-01387-f003:**
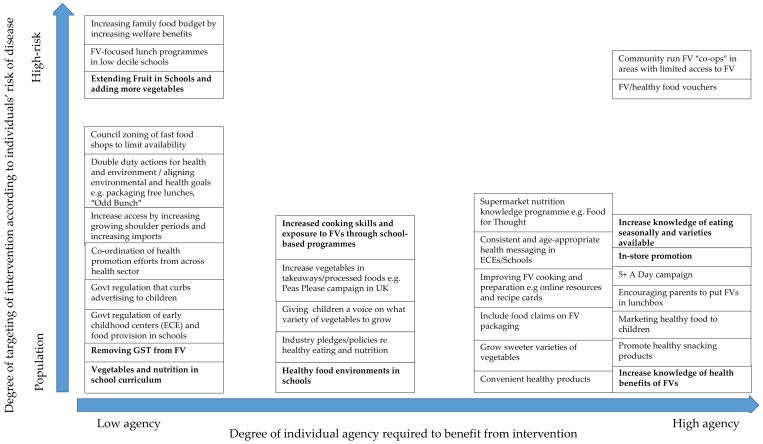
Public health interventions framework showing the likely effectiveness and impact of the interventions proposed on socioeconomic inequalities. Adams et al. propose that interventions in the top left hand corner will likely be the most equitable and effective [10]. Interventions in bold are those mentioned in four or more interviews.

**Table 1 ijerph-16-01387-t001:** Description of participants in the cognitive mapping interviews.

Interview Code	Date and Location	Sector	Organisation Description	Role
109	12.04.18 Interviewee’s Workplace	Government	Local Government	Sustainability & Resilience Advisor
103	05.04.18 Interviewee’s Workplace	Government	Central Government	Portfolio Manager
113	03.05.18 Video conference	Government	Central Government	Retail Sales Advisor
113				Senior Advisor Food Claims
107	10.04.18 Interviewee’s Workplace		Health Promotion	Nutrition & Activity Advisor
110	13.04.18 Interviewee’s Workplace		Health Promotion	Marketing Manager
111	16.04.18 Videoconference		Health Promotion	Māori Nutrition Kaiārahi
112	04.05.18 Interviewee’s Workplace		Health Promotion	Education Setting Manager
112				Programme Manager
112				Pacific Education
112				Food & Nutrition Manager
112				National Nutrition Advisor
101	29.03.18 University of Auckland		Social policy	Chief executive
106	10.04.18 Interviewee’s Workplace	Private	Research	Director
105	06.04.18 Interviewee’s Workplace	Produce Industry	Grower & Wholesaler	Executive GM of NZ Produce
115	26.04.18 Interviewee’s Workplace	Produce Industry	Grower	Marketing Manager
115				Owner
116	12.04.18 Interviewee’s Workplace	Produce Industry	Grower	Owner and Marketing Manager
118	14.05.18 Interviewee’s Workplace	Produce Industry	Grower	Owner
108	10.04.18 Interviewee’s Workplace	Produce Industry	Industry Body	Chief Executive
102	04.04.18 University of Auckland	Produce Industry	Health promotion	Education & Marketing Manager
104	06.04.18 Interviewee’s Workplace	Retail Sector	Supermarket	Head of Produce

NOG: Non-government organization.

**Table 2 ijerph-16-01387-t002:** Themes and subthemes from CMI interviews.

Theme	Subtheme
Price	Actual price
	Perceived price
Social context	Socioeconomic conditions
Food environment	Community Home School Retail Marketing/advertising Workplace
Time	Convenience Preparation
Skills and knowledge	Skills Knowledge
Food preferences and taste	Preference Taste
Built and natural environment	Weather Climate change Urbanisation Land use Growing practices
Government & business	Social welfare benefits Council Imports/exports Funding Industry Policy & Regulations

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
