# Peer review of "Systemic Barriers and Equitable Interventions to Improve Vegetable and Fruit Intake in Children: Interviews with National Food System Actors"

_ijerph, 2019, doi:10.3390/ijerph16081387_

Round 1

Reviewer 1 Report

Please kindly address the comments

Author Response

Procedures

2.2 Please kindly outline the criteria for selecting CMI participants?? Was it a requirement for them to have children aged 2-14years who consume FV? Or was it required that they must be females/males who are involved in food preparation for the children?? Were they supposed to be in management positions in institutions which distribute or sell FV? Please look at line 177-178 “the other five sent the invitation on to colleagues they believed would be suitable”, hence the need to clarify in section 2.2 the criteria which determined the suitability of participation. 

Response: see line 114. We have clarified that the criteria for selection was “based on employment in a prominent role in one of four sectors of the New Zealand food system: government/regulatory, produce industry, retail industry or public health/nutrition promotion.”

Please clearly explain you sampling technique, who exactly did you target (NGO, private etc), why these organisations?? Are they distributors or sellers of FV, How many did you target (sample size), how many participated (final sample size, 16 participants in this case). Justify why the final sample size is enough for you to draw conclusions considering the methodology you are using, is this a representative sample for your target population?? 

Response: As explained above, we aimed to include at least 2 representatives from four sectors, including government/regulatory, produce industry, retail industry or public health/nutrition promotion. Within these sectors we approached 17 organizations and all of these took part, with the exception of one supermarket chain. See line 204 and the reworked procedures paragraph from line 113.

In a table format, please indicate when and where each interview was conducted. 

Response: We have now added this to Table 1 and added a summary in text. Line 205-207.

Please also emphasize that this is a qualitative approach 

Response: added in line 80 and the abstract (line 25).

Results

Line 178: Please Delete “One organization (a large supermarket retailer) did not 178 respond to multiple invitations to participate” no need to mention here.

Response: We have modified this wording. 

This section should present results. Please consider moving line 176 to 179 to section 2.2 as this sounds like a method which was used to recruit, this will address bullet point no. 2 but please expand further as indicated in bullet point no. 2 

Response: Moved to procedures as suggested - see new lines 118-133 and 204-209.

Please rephrase Table.1 caption, confusing as it stands 

Response: Reworded as “Description of participants in the cognitive mapping interviews”

This line “In total, 16 interviews were held with 22 participants 179 from 5 different sectors of the food system (Table 1)” can be rephrased to “Table 1 shows………………..” cause already your 2.2 section would have covered the number of interviews. 

Response: Reworded as “Table 1 contains details of each interview and the participants’ roles and organisations.”

 Figure 2; please make sure font size and type is similar to rest of the text if possible.

Response: Corrected the font type but the size is difficult to correct in our software programme. Please let us know if this is still a problem.

 [Add tables to summarise the results] 

Response: Reviewer 2 requested for subheadings to be added to this section which we have done, and we believe this may be preferable to summary tables (which would duplicate information in the text). Please consider if the new subheadings (level 3) address your concerns.

Figure 3, font size not similar to rest of document 

Response: Unfortunately we have not been able to correct this. Please let us know if this will be an issue for readers.

Discussion

Delete “consisting of 16 interviews with 22 national… growers)”. This is already known. 

Response: Corrected as suggested.

Conclusion

631-633- Great, this should be added to your procedures, to add clarity to the reader 

Response: Lines 631-633 in the original manuscript were: A broad range of participants for this research was achieved, however, it would have been preferable to include both major NZ supermarket chains and an independent FV retailer, and possibly processed food manufacturers.

We have retained this sentence in the limitations section, replacing ‘possibly’ with ‘also’, but agree with your suggestion to add to the procedures about our attempt to include both supermarket chains. This is now line 115-117. The idea of interviewing an independent FV retailer and processed food manufacturers was made in hindsight after some participants mentioned that there was more they could be doing to improve FV intake.

Delete repetitions- Author contributions, funding, conflict of interest etc, Please make sure this is not repeated as it currently appears before and after the conclusion

Response: Corrected

Reviewer 2 Report

Relevant theme with a differentiated and innovative methodological approach.

Results: I suggest  in the description of interview participants explain how occurred the participation of 22 individuals in 16 interviews. Lines 176-180

There is a great deal of transcript of respondents' responses. To facilitate the reader's understanding, I suggest grouping and identifying respondents' responses by topic using subitems.

Discussion: Considering the barriers related to the intake of fruits and vegetables pointed out by the interviewees, it is important to bring the discussion of food and nutritional security in the article.

Author Response

Results: I suggest  in the description of interview participants explain how occurred the participation of 22 individuals in 16 interviews. Lines 176-180

Response: See amended line 190-191.

There is a great deal of transcript of respondents' responses. To facilitate the reader's understanding, I suggest grouping and identifying respondents' responses by topic using subitems.

Response: We have added a third level of headings in section 3.4 (barriers) and 3.5 (solutions). We also removed the last paragraph of barriers and multiple quotes to shorten and make it easier for readers to follow.

Discussion: Considering the barriers related to the intake of fruits and vegetables pointed out by the interviewees, it is important to bring the discussion of food and nutritional security in the article.

Response: We have addressed household food insecurity in section 4.3. Please check if this section addresses your concerns.

Reviewer 3 Report

Thank you for the opportunity to review this paper, which explores structural barriers to fruit and vegetable consumption among children in New Zealand, using a novel qualitative methodology and a less commonly included set of stakeholders. Strengths include the engaging interview process whereby individuals could comment on maps in real time, as well as on their complete interview maps to affirm or suggest edits; and a unique method of mapping themes. The authors take a systems-approach to their questions, yet many of the reflections shared by interviewees comment on barriers of an individual nature. The paper would be strengthened by clarifying a few specific items (below) and more generally shortening many of the sections, in order to arrive more immediately to solutions proposed.

1.      The abstract lists interviews with “23 national actors” in the food system but the body of the text repeatedly lists 22.

2.      While the introduction first presents consumption data relevant to New Zealand, the research describing determinants of consumption are international reviews. It would be interesting to know if there are findings in NZ-specific studies regarding determinants of consumption that would inform the study’s geographic focus.

3.      Page 4, Line 141: What was the “small gift” provided to interviewees? Was it monetary?

4.      Page 5, line 176: I assume with 16 individuals and 22 people that some interviews happened in groups. Please comment on this – were there differences in how group interviews were conducted? Were individuals in a group interview all representing the same organization and/or sector or was it a mix? Pros and cons of group vs individual interviews and do you think the make-up influenced the discussion? The authors mention at the paper’s conclusion that Group Model Building is a usual (and preferred) approach but was not possible due to logistical challenges of interviewee schedules – but it appears they did a mix of both. This should be addressed earlier in methods and the possible effects on results should be discussed.

5.      The map is great, with a few suggestions: In top third, a typo “commercial imact” – do you mean impact? Also, there are many words broken up between lines in unconventional ways. Would be easier to read if you could fix this formatting.

6.      Page 7, line 22: “lots of unprocessed foods [create] micronutrient deficiencies in children…” – shouldn’t this quote read “lots of processed foods” instead? That would make more sense intuitively but I realize it’s a quote hence the question.

7.      A few items that are clear oversights in proofing the text: Page 17 line 579 there’s a note that appears to be to the coauthors that reads “[Add about Sally’s research]”. Also, on Page 18, the acknowledgements, funding, conflicts of interest, and author contributions section appear copied text from author guidelines.

8.      Overall, the sections are long, making the paper as a whole hard to follow at times. For each subtheme there are sometimes 3-4 long quotations, and the reader may get lost in the flow of the themes/results. It would be worth reviewing these especially long sections and keeping to 1-2 very strong quotes per sub-item, then focusing the heart of outcomes on the solutions discussed.

Author Response

1.    The abstract lists interviews with “23 national actors” in the food system but the body of the text repeatedly lists 22.

Response: Thank you for noticing this typo. Corrected in the abstract.

2.    While the introduction first presents consumption data relevant to New Zealand, the research describing determinants of consumption are international reviews. It would be interesting to know if there are findings in NZ-specific studies regarding determinants of consumption that would inform the study’s geographic focus.

Response: There have not been any NZ-specific studies on determinants of children’s fruit and vegetable intake that we are aware, beyond what is written in lines 48-51 about the ethnic/socio-demographic differences.

3.    Page 4, Line 141: What was the “small gift” provided to interviewees? Was it monetary?

Response: We have clarified in line 148 (a bottle of local olive oil)

4.    Page 5, line 176: I assume with 16 individuals and 22 people that some interviews happened in groups. Please comment on this – were there differences in how group interviews were conducted? Were individuals in a group interview all representing the same organization and/or sector or was it a mix? Pros and cons of group vs individual interviews and do you think the make-up influenced the discussion? The authors mention at the paper’s conclusion that Group Model Building is a usual (and preferred) approach but was not possible due to logistical challenges of interviewee schedules – but it appears they did a mix of both. This should be addressed earlier in methods and the possible effects on results should be discussed.

Response: Clarified on page 5, line 204 onwards.

5.    The map is great, with a few suggestions: In top third, a typo “commercial imact” – do you mean impact? Also, there are many words broken up between lines in unconventional ways. Would be easier to read if you could fix this formatting.

Response: Corrected ‘impact’ and the broken lines

6.    Page 7, line 22: “lots of unprocessed foods [create] micronutrient deficiencies in children…” – shouldn’t this quote read “lots of processed foods” instead? That would make more sense intuitively but I realize it’s a quote hence the question.

Response: Thank you for spotting this. Was just a typo, corrected now.

7.    A few items that are clear oversights in proofing the text: Page 17 line 579 there’s a note that appears to be to the coauthors that reads “[Add about Sally’s research]”. Also, on Page 18, the acknowledgements, funding, conflicts of interest, and author contributions section appear copied text from author guidelines.

Response: Thank you – both corrected now.

8.      Overall, the sections are long, making the paper as a whole hard to follow at times. For each subtheme there are sometimes 3-4 long quotations, and the reader may get lost in the flow of the themes/results. It would be worth reviewing these especially long sections and keeping to 1-2 very strong quotes per sub-item, then focusing the heart of outcomes on the solutions discussed.

Response: Agreed. We have gone through the manuscript again deleting quotes when there were three given in a section, and we have deleted a paragraph (3.4.9). The new sub-headings added to sections 3.4 and 3.5 should also assist readers.

Int. J. Environ. Res. Public Health EISSN 1660-4601 Published by MDPI AG, Basel, Switzerland RSS E-Mail Table of Contents Alert
Back to Top